# Electrical Stimulation Increases Axonal Growth from Dorsal Root Ganglia Co-Cultured with Schwann Cells in Highly Aligned PLA-PPy-Au Microfiber Substrates

**DOI:** 10.3390/ijms23126362

**Published:** 2022-06-07

**Authors:** Fernando Gisbert Roca, Sara Serrano Requena, Manuel Monleón Pradas, Cristina Martínez-Ramos

**Affiliations:** 1Center for Biomaterials and Tissue Engineering, Universitat Politècnica de València, 46022 Valencia, Spain; saserre@etsii.upv.es (S.S.R.); mmonleon@ter.upv.es (M.M.P.); cris_mr_1980@hotmail.com (C.M.-R.); 2CIBER-BBN, Biomedical Research Networking Center in Bioengineering Biomaterials and Nanomedicine, 28029 Madrid, Spain; 3Unitat Predepartamental de Medicina, Universitat Jaume I, 12071 Castellón de la Plana, Spain

**Keywords:** electrical stimulation, alternating current, axonal growth, aligned substrates, polylactic acid, polypyrrole, Schwann cells

## Abstract

Nerve regeneration is a slow process that needs to be guided for distances greater than 5 mm. For this reason, different strategies are being studied to guide axonal growth and accelerate the axonal growth rate. In this study, we employ an electroconductive fibrillar substrate that is able to topographically guide axonal growth while accelerating the axonal growth rate when subjected to an exogenous electric field. Dorsal root ganglia were seeded in co-culture with Schwann cells on a substrate of polylactic acid microfibers coated with the electroconductive polymer polypyrrole, adding gold microfibers to increase its electrical conductivity. The substrate is capable of guiding axonal growth in a highly aligned manner and, when subjected to an electrical stimulation, an improvement in axonal growth is observed. As a result, an increase in the maximum length of the axons of 19.2% and an increase in the area occupied by the axons of 40% were obtained. In addition, an upregulation of the genes related to axon guidance, axogenesis, Schwann cells, proliferation and neurotrophins was observed for the electrically stimulated group. Therefore, our device is a good candidate for nerve regeneration therapies.

## 1. Introduction

Nervous system regeneration involves great challenges due to its scarce intrinsic repair and needs to be guided when the injury is longer than 5 mm to achieve a successful regeneration [1]. For this reason, substrates with different topographic cues have been studied, with the aim of guiding axonal growth and thus preventing axons from growing randomly [2,3]. One of these strategies consists of the use of highly aligned fibrillar substrates, which guide axonal growth in the direction of the fibers [2,3,4,5,6,7]. In this study we have employed a substrate based on highly aligned polylactic acid (PLA) microfibers for this purpose. PLA is a synthetic polymer that has been proven to be biocompatible, biodegradable and bioabsorbable in the human body [8,9,10,11]. It has been widely employed in tissue engineering as it safely degrades along the same metabolic pathway as lactic acid [12], being approved by the U.S. Food and Drug Administration (FDA) [13,14,15]. PLA substrates based on aligned microfibers have been shown to highly support neural stem cell culture and improve neurite growth, guiding axonal growth in an aligned direction [16]. In addition, an increase in axonal growth and extension has been previously observed when microfiber-based substrates are employed instead of nanofiber-based ones [17]. Due to these properties of PLA and the positively developed studies in the field of nerve regeneration, this material has been chosen for the present work.

One of the major problems with nerve regeneration is that it is a very slow process, with an average rate of axonal growth of 1 mm/day that cannot sustain itself for more than 12 months [18,19,20]. For this reason, different strategies have been studied to accelerate axonal growth. One of these strategies is electrical stimulation employing electrically conductive substrates, which have shown to increase the axonal extension of neurons and to induce a higher release of neurotrophins by glial cells when subjected to an exogenous electric field [21,22], supporting the idea that electrical stimulation plays a very important role in promoting nerve regeneration. For this reason, here the PLA microfibers are coated with the electroconductive polymer polypyrrole (PPy). PPy presents a good biocompatibility both in vitro and in vivo, good chemical stability, long-term ambient stability and high electrical conductivity under physiological conditions [23]. In addition, it can be easily synthesized by chemical or electrochemical polymerization with different porosities and surfaces [24,25,26,27], which makes it an ideal material for biomedical applications [23,28]. PPy has been widely employed in biomedical applications, particularly in nerve tissue engineering, exploiting its good biocompatibility and high electrical conductivity [4,24,27,29,30,31,32]. PPy has shown to improve the axonal extension of neurons, favour cell adhesion and induce a greater secretion of growth factors by glial cells when subjected to an exogenous electric field [6,33,34,35]. Another electrically conductive material that is employed in this work is gold (Au). It is chemically inert with low chemisorption capacity [36,37]. The use of gold as a biomaterial for nerve regeneration has recently increased, usually conjugated with other materials [38,39]. Au has been employed for the stimulation and modulation of neural activity with different applications, such as improving cell growth [38]. Furthermore, Au presents a good biocompatibility when in contact with body fluids and tissues, as well as adequate characteristics to avoid immunogenic reactions [36,37].

In this study, PLA-PPy-Au microfiber substrates were employed for the cell culture of Schwann cells (SCs) and dorsal root ganglia (DRG) to demonstrate the regenerative effects of electrical stimulation. A key role in axonal regeneration is played by glial nerve cells, since they supply growth factors and biochemical signals that are necessary for axonal growth [40,41,42,43,44,45]. Among them, SCs stand out because they are essential for neuronal survival and axonal regeneration in the peripheral nervous system [46,47,48]. SCs are responsible of maintaining the myelin sheath around the axons and secrete different neurotrophic factors such as nerve growth factor (NGF), brain-derived neurotrophic factor (BDNF) and neurotrophin-3 (NT-3), among others [46,47,48]. These neurotrophic factors are essential to create a biochemical environment that favours axonal growth. Furthermore, in the case of injury, SCs promote axonal regeneration by forming cell columns, called Büngner bands, that guide the regenerating axons [49,50]. SCs also release cytokines, such as LIF and IL-6, that promote the survival of neurons [46].

The aim of this work is to obtain highly aligned electroconductive PLA-PPy-Au microfiber substrates and characterize their physicochemical and electric properties, as well as their biological validity, to increase axonal growth and sprouting when subjected to an exogenous electric field. For this, a bioreactor was designed and manufactured to arrange the PLA-PPy-Au microfiber substrate at the bottom of the culture wells with a highly aligned and flat arrangement. To take advantage of the electroconductivity of PLA-PPy-Au substrates, the bioreactor was designed in such a way that it was possible to establish an electrical contact between the electrodes and the substrate outside the culture well, thus avoiding the direct contact of the electrodes with the culture medium. With this starting point, the goal of this work is to study if, by means of the electrical stimulation of the developed electroconductive substrates, a beneficial effect can be obtained on the axonal growth and sprouting of the neurons present in the DRG body, exploring its suitability as an electroconductive bioactive substrate for nerve regeneration applications.

## 2. Results and Discussion

### 2.1. Stability and Degradation Study of PLA-PPy Microfibers

In this work, substrates made of highly aligned PLA microfibers were coated with the electrically conductive polymer polypyrrole (PPy) via in situ polymerization with a mass fraction of PPy of 3.5% (Appendix A). In previous studies [17,51], it was observed by scanning electron microscopy that these PLA-PPy substrates showed a continuous and homogeneous PPy coating with a fine grain texture. Only small PPy aggregates were observed and most of them disappeared with repeated washing, so this coating protocol with PPy was applied.

To verify the stability of the electrical conductivity of the PLA-PPy microfibers, the conductivity of the samples stored under atmospheric conditions and under vacuum conditions was monitored for 35 days. As can be seen in Figure 1, a loss of conductivity of the PLA-PPy microfibers was observed in both conditions. However, a significant difference was observed between both storage conditions. The samples stored under atmospheric conditions suffered a greater loss of conductivity, losing 30 ± 4% of conductivity after 35 days, while samples stored under vacuum conditions only suffered a 11 ± 2% loss of conductivity after 35 days, which is a small loss in a relatively long time. These results indicate that storing the PLA-PPy microfibers in vacuum conditions protects them against environmental agents that produce oxidation effects that affect the conductive properties of the PPy coating. Therefore, the PLA-PPy microfibers employed in the following experiments were stored under vacuum conditions to increase the stability of the PPy coating. However, it was decided to reduce the storage time of the samples as much as possible, never exceeding 5 days. Furthermore, the short duration of the experiments (4 days) meant that the conductivity loss was only around 5%.

By means of a Fourier transform infrared spectroscopy (FTIR) analysis, the characteristic peaks of pure PPy, PLA microfibers and PLA-PPy microfibers were characterized and analysed (Figure 2). In addition, samples of PLA-PPy microfibers degraded in atmospheric conditions for 180 days and degraded in PB 0.1 M and 37 °C for 90 days were also studied. At 985 cm^−1^, a characteristic peak of the PPy dopant element (pTS) is observed for all the PLA-PPy samples, which corresponds to the stretching vibration of the S=O bonds of the −SO_3_^−^ group [52,53]. The peak is attenuated when there is a degradation process, being more attenuated for the group degraded in PB 0.1 M and 37 °C than for the group degraded in atmospheric conditions. Around 1546 cm^−1^, a characteristic PPy peak corresponding to the fundamental vibration of the PPy ring is observed for PPy and all PLA-PPy samples [54]. For the PLA-PPy samples degraded in atmospheric conditions, this peak appears slightly attenuated, which could be related to a degradation process in atmospheric conditions that has affected the stability of PPy. For the PLA-PPy samples degraded in PB 0.1 M and 37 °C, this peak is practically inexistent, indicating that the PPy has been completely affected by the treatment carried out. Finally, for the degraded groups, two peaks that are not present in the other groups are observed. These peaks correspond to a wavelength around 2849 cm^−1^ and 2917 cm^−1^ and show the bending and tension of the aliphatic C-H group [55]. These groups originate after a rupture of the pyrrole rings [55], which leads us to think that this rupture is due to the degradation of the PPy produced in the degraded samples under the different conditions.

Through a thermogravimetric analysis (TGA), the stability of PLA-PPy samples after the different degradation processes (180 days in atmospheric conditions and 90 days in PB 0.1 M and 37 °C) was studied. As can be observed in Figure 3, in the curve that corresponds to the PPy group, there is a loss of mass between 50 °C and 100 °C, which is mainly due to the evolution of the pTS dopant adsorbed on the sample and to the evaporation of H_2_O [56]. However, this weight loss is not observed for the samples composed of both materials (PLA-PPy) since the mass fraction of PPy in the samples is only around 3.5%. Regarding the mass loss of the PLA-PPy samples between 300 °C and 400 °C, it is due to the degradation of the polymer chain. This thermal degradation of the polymer chain occurs at a lower temperature when the material is degraded, which is why the samples degraded in atmospheric conditions and in PB 0.1 M and 37 °C have a thermal degradation of the polymer chain that occurs at a lower temperature than for the PLA-PPy non-degraded samples. This effect is more pronounced for the case of samples degraded in PB 0.1 M and 37 °C, since it is a more aggressive degradation, being produced at a lower temperature than for the PLA group. This is produced because the PPy coating that acted as a protective layer for the non-degraded PLA-PPy samples has been lost, since PPy degrades later than PLA, but also because the PB 0.1 M has caused a hydrolytic degradation of the PLA polymer chain. Therefore, after the treatment of PLA-PPy microfibers with PB 0.1 M and 37 °C, both materials (PLA and PPy) are affected by the degradation process. In the case of PLA-PPy fibers degraded in atmospheric conditions, there was only a partial degradation of the PPy coating and, therefore, its thermal degradation curve occurs later than the one corresponding to the PLA group. This allows us to conclude that the degradation in atmospheric conditions mainly affects the PPy coating, while the degradation in PB 0.1 M and 37 °C affects both the PLA microfibers and the PPy coating.

It is important to highlight that this is an in vitro study with ideal conditions. Regarding the in vivo situation after its implantation, the regeneration of nerves takes several months and we expect a loss of conductivity of PPy in this situation, but the PPy performance should be effective at least during the first weeks that are the more critical ones. Despite this should be further studied in future experiments, many groups have already reported the biodegradation properties of PLA-PPy blends [4,57,58,59]. PLA-PPy composites showed a good electroconductivity for long time in a simulated physiological solution (15% of the conductivity remained after 1000 h) and the in vivo biodegradation of the composites showed the non-toxicity of degradation products [57,58,59,60]. Therefore, we expect that the in vivo degradation of the PLA-PPy substrates will not lead to a problematic immune response. In addition, we have proved that the PPy coverage does not prevent the degradation of PLA microfibers, which is an advantage for the in vivo application since the PLA-PPy microfibers will be satisfactorily resorbed.

Regarding gold microfibers, gold is considered chemically inert with a high stability and its surface shows low chemisorption capacity [37]. This causes gold to exhibit low chemical reactivity, little catalytic activity, high stability against corrosion and oxidation [36,61,62]. Therefore, clinically, gold presents biocompatibility when in contact with body fluids and tissues, adequate characteristics to avoid immunogenic reactions [36,37]. Furthermore, as it is a metal, it has excellent conductivity and has been used in electrodes for electrical neurostimulation [63,64]. Gold is highly stable in vivo, and its conductivity should not vary during the time needed for nerve regeneration. In addition, on our study gold is not a coating of the PLA-PPy microfibers, but is incorporated in the form of independent microfibers, so there should be no chemical interaction between PPy and gold microfibers. For these reasons, the stability and degradation of gold microfibers has not been studied.

### 2.2. Bioreactor for the Electrical Stimulation

Our work presents various novelties within the field of nervous tissue engineering. On the one hand, the diameter of the electroconductive fibers is 10 microns, which is not the usual size used in this type of studies. Since the fiber substrates obtained by means of the electrospinning technique are normally used, nanofibers with a diameter below 1 micron are commonly employed [4,6]. We previously observed that the use of microfiber substrates led to an increased axonal extension and axonal sprouting than the use of nanofiber substrates, since on microfibers the cell mobility was much less hindered than on nanofibers, and the cell–cell interactions were favoured, resulting in a higher motility of SCs that accelerated their migration speed from the DRG body, which is directly related to the axonal extension [17]. In addition, the PPy coating of the PLA microfibers resulted in an additional beneficial effect on axonal growth, possibly produced by the greater surface roughness that favoured the adhesion of DRG [17].On the other hand, most studies of electrical stimulation of DRG or isolated neurons apply stimulation by introducing electrodes into the culture medium, commonly using non-electroconductive substrates, since the current circulates through the culture medium and not through the culture substrate [65,66,67]. Furthermore, the few studies that apply the electrical stimulation through an electroconductive substrate do not employ microfibers [35,68,69]. Therefore, the fact that we use a substrate based on electroconductive microfibers that is connected outside the culture well (without medium-electrode contact)—so that the current circulates mainly through the culture substrate and not through the culture medium—represents a novelty in this field of study. In addition, the combination of PPy-coated microfibers with gold microfibers is also a novelty and, with this disposition, an electric field around the gold microfibers is created, generating an additional type of stimulation.

For the electrical stimulation of the SCs and DRG, an 8-well bioreactor was designed and manufactured, were the PLA-PPy-Au substrates were placed at the bottom of the wells and on which the SCs and the DRG were seeded. As can be observed in Figure 4, an 8-well Millicell^®^ EZ slide was employed, thanks to the fact that it can be disassembled and assembled in its different components, allowing the incorporation of PLA-PPy-Au microfiber substrates between the slide and the wells. The substrates were produced with a length greater than that of the slide in order to allow the contact of the electrodes with the electroconductive substrate outside the slide chamber, thus avoiding introducing the electrodes into the culture medium. This was done with the aim of maximizing the circulating current through the substrate and minimizing the circulating current through the culture medium.

For the electrical stimulation, it was decided to establish the applied current as the main parameter to control. Despite the setup, it was observed that if stimulation with direct current (DC) was applied for a period of several hours, an accumulation of the ionic charges of the culture medium led to a variation of its pH, turning into a yellow colour (Appendix A). This implied that an important part of the current was circulating through the culture medium. This was because the conductivity of the culture medium and that of the PLA-PPy substrate were similar: 1.70 ± 0.02 S/m and 1.4 ± 0.2 S/m, respectively. Despite being two different types of conduction (ionic for the culture medium versus electronic for the PLA-PPy substrate), this did not prevent the diversion of part of the current through the culture medium. For this reason, it was necessary to increase the conductivity of the substrates. One option could have been to increase the mass fraction of PPy present in the microfibers, but this implied the appearance of large PPy aggregates that would make it difficult for the axons to extend on the surface of the microfibers [56]. Furthermore, the increase in conductivity was not large enough to prevent current flow through the culture medium. For this reason, it was decided to introduce three gold microfibers of 25 µm in diameter below the PLA-PPy substrate. Thus, it was possible to increase the conductivity of the substrate by several orders of magnitude, up to 2.4 × 10^5^ ± 0.1 S/m. When gold microfibers were employed, no colour change was observed in the culture medium despite using DC (Appendix A), so the circulating current through the culture medium had been greatly reduced.

However, this does not imply the lack of usefulness of PLA-PPy substrates since DRG do not adhere properly to a bare gold substrate, as can be observed in Appendix A, where DRG seeded on top of gold substrates show a poor adherence to the substrate so there is no axonal extension after 4 days of cell culture. We attribute this to the fact that the electrochemical nature of bare gold is very neutral, which is why it does not favour the adsorption of proteins on gold during the pre-conditioning process of the substrate and, therefore, greatly hinders the adhesion of DRG to gold. Anchorage-dependent cells make integrin-mediated cell adhesion, which has no affinity for bare gold, so the gold surface must be modified—for example, with thiol-functionalized RGD peptide—in order to achieve cell adhesion to gold [70]. In addition, the high autofluorescence of gold prevents the realization of fluorescence images. For these reasons, it was decided to place the gold microfibers below the PLA-PPy substrate, so the SCs and DRG only had direct contact with the PLA-PPy microfibers. In addition, the use of PLA-PPy microfibers presents more advantages in front of using non-conductive PLA microfibers. On the one hand, part of the current will circulate through the PLA-PPy microfibers, although it will be small compared to the current that circulates through the gold microfibers. On the other hand, the electric field that is created around the gold microfibers would be much more shielded if non-conductive PLA microfibers were used instead of PLA-PPy microfibers.

In addition, to minimize the effect of the accumulation of charges that could occur with part of the current circulating through the culture medium, even if it is low, it was decided to switch from DC to alternating current (AC). Another possible solution could have been to continue using DC but including salt bridges to maintain the charges balance. However, the developed bioreactor was an 8-well system, and this implementation was very complex to perform. In addition, several studies had previously successfully applied stimulation with alternating current to stimulate axonal growth [35,66,69,71,72], so it was decided to apply this type of current for the electrical stimulation. Regarding the future in vivo application of the electroconductive substrate, it would be necessary to study the need to include gold microfibers according to the conductivities of the surrounding tissue but, if necessary, gold is a biocompatible material that is already used in various biomedical applications in the nervous system [62,73,74,75]. Furthermore, since gold microfibers represent a very small percentage of the total number of microfibers (0.5%), and their diameter is similar to the diameter of PLA-PPy microfibers (25 µm vs. 10 µm), there is not a significant stiffening of the system.

To increase the number of stimulated samples in the same experiment, three devices were connected in parallel, as shown in Figure 5. In this way, the current will be distributed uniformly throughout the six stimulated lanes, since it is the parameter that we wanted to set. As can be observed in the diagram, an ammeter was connected in series to set the circulating current, being six times the current that was desired to circulate through each PLA-PPy-Au lane. A potentiometer connected in series was also introduced to increase the electrical resistance of the system, since the incorporation of gold microfibers made the resistance of the system too low for the voltage generator that was used. This AC voltage generator was set to the desired frequency with a sinusoidal signal. In turn, the voltage generator was connected to a 24-h timer to set the hours that the system should be connected.

### 2.3. Electrical Stimulation of DRG and SC

The aim of this study is the electrical stimulation of entire DRG (not just isolated neurons) in a co-culture with pre-seeded SCs, so it is a co-stimulation of DRG and pre-seeded SCs. The type of current typically used in the stimulation of neurons or DRG is direct current (DC), introducing salt bridges to minimize the exposure of cells to changes in pH due to the accumulation of ionic charges in the culture medium [65,67]. In this study, we applied alternating current (AC) in order to avoid the accumulation of ionic charges of the culture medium and thus avoided having to use salt bridges. AC also has been applied in several studies for the electrical stimulation of axonal growth with good results [35,71].

As can be observed in Table 1, different stimulation conditions were studied, varying the applied electrical current and the frequency of the sinusoidal signal, keeping constant the stimulation time, the culture time, the number of pre-seeded SCs and the origin and age of the DRG. As a result, when a frequency of 250 Hz and a current of 3 mA was applied (experiment #1), an adverse effect of the stimulation was obtained, after which it was decided to lower the applied current to 1 mA without changing the frequency (experiment #2), which also gave rise to an adverse effect. In view of these results, it was decided to reduce both parameters, lowering the current to 0.5 mA and the frequency to 50 Hz (experiment #3), after which no effect of stimulation was observed on the growth of DRG axons, neither positive nor negative. Finally, the current was maintained in 0.5 mA and the frequency was lowered to 3 Hz (experiment #4), which gave rise to a positive effect on axonal growth, showing an increased axonal extension and axonal sprouting for the electrically stimulated DRG, compared with the non-stimulated ones. Several studies of neurostimulation of DRG have observed that low frequencies (<20 Hz) have a beneficial effect on pain relief, inducing the dorsal horn of the spinal cord inhibition via the activation of low threshold mechanoreceptors and the release of endorphins and dynorphins [76,77,78,79]. Therefore, we could say that the applied stimulation frequency of 3 Hz is within the physiological range of DRG activation. Regarding the applied current, a current of 1 mA has been applied also for neural activation during DRG stimulation [76], so the applied frequency of 0.5 mA is also in the physiological range of DRG activation. The results of experiment #4 are explained below.

In previous studies [17,51], confocal images of the SCs cultured on PLA-PPy microfiber substrates showed an effective adherence and proliferation of the cells together with a high cell alignment. Immunofluorescence images of the DRG cultured on PLA-PPy microfiber substrates also showed a highly aligned axonal extension, without a detrimental effect when the PPy coating was present [17]. In this work, as can be observed in Figure 6, control and electrically stimulated samples were stained with Neuron-specific β III Tubulin antibody in green colour (Figure 6A,D, respectively), with S100β antibody in red colour (Figure 6B,E, respectively) and DAPI in blue colour (Figure 6C,F, respectively). As can be observed in Figure 6A,D, the staining with Neuron-specific β III Tubulin antibody was employed to test the differences in axonal extension and sprouting. In the centre of the images, the DRG body can be observed, which contains the neuronal somas and from which the axons arise in a highly aligned way thanks to the highly aligned PLA-PPy microfiber substrate. The staining with S100β antibody was employed to observe the SCs, both the pre-seeded ones and those that migrated from the DRG body. It has been previously observed [51,80,81] that, when there are not pre-seeded SCs, the SCs that migrated from the DRG body are only present near to the axons, with no SCs in areas where axons are not present. Therefore, observing SCs at the ends of the microfiber lane, far away from the axons, indicates the presence of the pre-seeded SCs. This can be also observed with the DAPI staining, since the nuclei of the pre-seeded SCs can be observed far away from the axons.

As can be observed, both the maximum length of the axons and the area occupied by them is greater for the stimulated DRG (Figure 6D) than for the non-stimulated DRG (Figure 6A). These two parameters were quantified as explained in Section 3.13 (quantification of axonal extension and axonal sprouting) with a total amount of 16 samples per group (*n* = 16) and the results are presented in Figure 6G,H. As can be observed in Figure 6G, the quantification of the maximum length of axons shows a statistically significant increase for the stimulated samples. Since the axons grow at both sides of the DRG body, this parameter was measured as the maximum, the sum and the average of both sides in order to study the symmetry of axonal growth. As a result, an increase in the stimulated samples of 17.9% (2.8 ± 0.1 mm vs. 3.3 ± 0.1 mm), 19.2% (5.2 ± 0.2 mm vs. 6.2 ± 0.2 mm) and 19.2% (2.6 ± 0.1 mm vs 3.1 ± 0.1 mm) was observed, respectively. This proved that the measures were constant and symmetrical since similar results were obtained for the three cases. Likewise, the total area covered by the axons was also quantified, which is indicative of the degree of axonal sprouting. As seen in Figure 6H, an increase in the area of axons of 40% (0.48 ± 0.04 mm^2^ vs. 0.67 ± 0.04 mm^2^) was obtained for the electrically stimulated DRG compared to the non-stimulated ones.

As can be observed in a detailed confocal image of an area far from the DRG body (Figure 7), the axons (in green colour) grow on the top of PLA-PPy microfibers in a highly directed manner (marked with *), both in the control group (Figure 7A) and in the stimulated group (Figure 7B). Likewise, the SCs are also observed (in red colour), which are also arranged on the top of PLA-PPy microfibers accompanying the axons. In addition, it is observed that the SCs have a highly elongated cytoskeleton in the direction of the microfibers (marked with arrows).

### 2.4. qRT-PCR Analysis after the Electrical Stimulation

As can be observed in Figure 8, the relative expression of the genes related with axon guidance and axonogenesis, Schwann cells, cell proliferation and neurotrophins was determined by real-time qPCR.

The axon guidance- and axonogenesis-related genes, the β III Tubulin, the growth associated protein 43 (GAP 43), the neurofilament middle chain (NF-M) and the neurofilament light chain (NF-L) were studied. Since all the proteins encoded by these genes are located in neurons, the chicken sequences of these genes were studied (see Table 2). The β III Tubulin protein is a microtubule element that plays an important role in normal neural development, regulating ligand interactions and microtubule formation. It has implications in neurogenesis and axon guidance and maintenance [82,83]. The GAP 43 is a protein related with the neuronal growth and plasticity. It is expressed in neuronal growth cones during axonal regeneration. It is a key component of the axon and the presynaptic terminal, and it is a widely used marker of axonal sprouting. It mediates axonal growth, branching, and pathfinding during development but also in axonal remodelling [84,85,86]. Neurofilaments are the most abundant cytoskeletal molecules in myelinated neurons. These proteins expressed in neurons play an important role in the intracellular transport of molecules throughout the axons and dendrites, contributing to the integrity of neuronal cytoarchitecture. They increase the diameter of neuronal axons, facilitating electrical conduction and neuronal signalling [87]. As can be observed in Figure 8A, an upregulation of all these genes was observed for the electrically stimulated group, being specially elevated for the GAP 43. This implies that the stimulated group presents an increased rate of neuronal growth.

The SC-related protein S100β, is considered as a glial marker protein, since it is very abundant in Schwann cells, astrocytes and oligodendrocytes, among others [88,89]. It is involved in cell proliferation, survival and differentiation as an intracellular regulator and as an extracellular signal [88]. It regulates cell shape, cell-to-cell communication, intracellular signal transduction, energy metabolism and cell growth, among other functions [89]. As can be observed in Figure 8A, a clear upregulation of S100β can be observed. Since most of the cells that migrate from the DRG body are SCs [80,81], this upregulation can be related to a higher proliferation, survival and cell-to-cell communication of SCs for the electrically stimulated group. This is of great importance for axonal growth since the support of SCs directly influences the guidance and growth of axons, by the release of neurotrophic factors and other signals that favour axonal growth. It is important to note that this result is related with the chicken SCs that migrate from the DRG body, not with the pre-seeded rat SCs.

The cell proliferation-related protein (PCNA) is a nuclear protein synthesized in the early G1 phase and in the S phase of the cell cycle. It is located in the nuclei of cells that undergo cell division and promotes DNA synthesis, as it is a cofactor for DNA polymerase delta. PCNA has a key function in the cell cycle regulation and DNA replication and also plays a role in other processes involving the cell genome [90,91,92]. As can be observed in Figure 8A, it is also upregulated for the electrically stimulated group. Since it is expected that most of the chicken cells present in the culture are SCs that migrated from the DRG body [80,81], this upregulation of PCNA can be related to a higher proliferation of SCs in the electrically stimulated group.

In addition, rat genes related to the neurotrophins BDNF (brain-derived neurotrophic factor) and NGF (nerve growth factor) were also studied to assess if the electrical stimulation had a beneficial effect on the release of neurotrophins from pre-seeded SCs. On the one hand, BDNF helps to support the survival of neurons, activating the growth and differentiation of new neurons [93,94]. It is one of the most active neurotrophins, helping to stimulate and control neurogenesis and playing a key role in normal neural development [95,96,97,98]. On the other hand, NGF is involved in the growth and maintenance of neurons, stimulating its proliferation and survival [99,100]. As can be observed in Figure 8B, both neurotrophins were upregulated for the electrically stimulated group, indicating an increased secretion of this neurotrophic factors by the pre-seeded rat SCs. Thus, this is another factor that may be of great importance in accelerating axonal growth.

Therefore, we have studied the main biomarkers that are related with axonal growth and regeneration: β III Tubulin related with neural development, GAP 43 related with growth cones during axonal regeneration and neurofilament related with the integrity of neuronal cytoarchitecture, which were more expressed for the electrically stimulated group. Furthermore, two of the main neurotrophic factors (BDNF and NGF) were studied, which were also more expressed for the electrically stimulated group. Since the electrically stimulated group has a higher expression of neuronal, SC, proliferation and neurotrophins biomarkers, we can conclude that this group has a stronger nerve regeneration potential. It is also important to highlight that all the chicken and rat genes that were studied were species-specific (including the GAPDH), so there is no expression at all when only DNA from the other species is present.

## 3. Materials and Methods

### 3.1. Preparation of PLA Microfiber Bundles

Highly aligned PLA microfiber bundles were obtained by grouping 600 PLA microfibers with a diameter of 10 μm each (AITEX Textile Research Institute, Alcoy, Spain). In order to maintain the alignment of the lane-shaped microfiber bundles, they were fastened using polycaprolactone (PCL) bands, which were placed in a solid state on both extremes of the lane and melted by temperature so that, once cooled, the microfibers were attached by the PCL bands.

### 3.2. Coating of PLA Microfibers with PPy

The PLA substrates were coated with the electrically conductive polymer polypyrrole (PPy) via in situ polymerization with a mass fraction of PPy of 3.5%. As a previous step, the substrates were immersed in deionized water under compression and a fixed vacuum was applied until they stopped floating and, therefore, the introduction of water inside the spaces between fibers was achieved, in order to obtain a homogeneous coating of all microfibers, not only the most superficial ones. Next, each substrate was put into a polypropylene tube with an aqueous solution of pyrrole monomer (Py, Sigma-Aldrich, 131709, Madrid, Spain) and sodium para-toluene sulfonate (pTS, Sigma-Aldrich, 152536, Madrid, Spain), followed by ultrasonication for 1 min in order to allow the microfibers to be saturated with the Py/pTS solution. The substrates were incubated with shaking at 4 °C for 1 h. The ratio between the substrate area (length × width) and the final volume of the Py/pTS aqueous solution was 0.6 cm^2^/mL. A concentration of 14 mM was used for both Py and pTS. Then, an aqueous solution of ferric chloride (FeCl_3_, Sigma-Aldrich, 157740, Madrid, Spain) was added and incubated with shaking at 4 °C for 24 h for the polymerization and deposition of PPy on the PLA substrates. The ratio between the substrate area (length × width) and the final volume of the FeCl_3_ aqueous solution was 0.6 cm^2^/mL. The concentration of FeCl_3_ was 38 mM. PPy-coated membranes were washed with deionized water with agitation for 10 min for three times and ultrasonicated for 30 min in deionized water three times. Finally, the membranes were dried in a desiccator with a fixed vacuum at 40 °C for 2 days.

### 3.3. Bioreactor for Electrical Stimulation

The Millicell^®^ EZ slide chamber (Merck Millipore, PEZGS0816, Madrid, Spain) was employed to develop a bioreactor that allowed the direct contact between the electrodes and the electroconductive substrate while avoiding the direct contact between the electrodes and the culture medium (Figure 4). This commercial device was chosen as it was completely removable and allowed the substrate to be placed at the bottom of the wells. To do this, the substrate was first placed between the slide and the wells and, once the slide was assembled, the ends of the microfibers remained outside the slide, available to connect to the electrodes. In order to increase the conductivity of the PLA-PPy substrate, gold microfibers with a diameter of 25 μm each (GoodFellow, AU00-WR-000121, Madrid, Spain) were used. These gold microfibers were placed below the PLA-PPy microfiber substrate and a total of three gold microfibers were arranged in parallel and equally spaced from each other (Figure 4A). The gold microfibers were placed below the PLA-PPy lane because their larger diameter and their poorly adherent surface hindered the adhesion of SCs and, especially, of the DRG, which did not adhere to the substrate if they were in direct contact with the gold microfibers. Once the microfiber substrate was introduced in the slide chamber, the contact between the slide and the wells was no longer perfectly sealed. Thus, to avoid leaks of the culture medium, a cellulose-based sealant was added, which, once the chamber was closed and the solvents were evaporated, exerted a sealing and adhesive role that prevented the leakage of culture medium. As can be observed in Figure 4B, in the final arrangement of the device one microfiber substrate or lane is placed in the centre of each of the two rows of the slide. These lanes protrude from the slide in order to proceed to the connection with the electrodes outside the wells, in order to avoid direct contact with the culture medium. Therefore, the device consists of two lanes connected in parallel and, therefore, the circulating current will be distributed approximately 50% for each lane.

### 3.4. Stability and Degradation of PLA-PPy Microfibers

To study the stability of the PPy coating of PLA microfibers, electrical conductivity measurements were made every 7 days for a total of 35 days. The samples were stored both under atmospheric conditions (*n* = 6) and vacuum conditions (*n* = 6) and the electrical characterization was performed according to Section 3.5, calculating the conductivity loss relative to the initial conductivity value (in %). To study the degradation of PPy-coated PLA microfibers, they were subjected to two different conditions. On the one hand, they were kept for 180 days under atmospheric conditions (*n* = 6), and on the other hand, they were kept submerged in phosphate buffer (PB) 0.1 M at 37 °C for 90 days (*n* = 6). After this time, the samples were studied by FTIR and TGA, as explained in Section 3.7 and Section 3.8.

### 3.5. Electrical Characterization

The electrical resistance of the different samples was measured at room temperature placing one electrode at each extreme of the sample, with a space between them (*d*) of 10 cm, and applying a constant voltage of 5 V. The circulating current was measured with a multimeter, and the apparent surface electrical resistance (*R*) was calculated applying the Ohm’s law. The average value was reported from five different samples (*n* = 5). Once the electrical resistance was obtained, it was divided by the distance at which the electrodes were placed when taking the measurement. After measuring the cross section of the substrates (*S*), the in-plane apparent DC conductivity ($\sigma_{DC}$) of the samples was obtained by Equation (1).
(1)$$\sigma_{DC}=\frac{d}{R\times S}$$

### 3.6. Field Emission Scanning Electron Microscopy (FESEM)

The morphological characterization of the surface of PLA-PPy microfibers was performed using a field-emission scanning electron microscope (FESEM; ULTRA 55, ZEISS, Oxford Instruments, Wiesbaden, Germany). The preparation of samples consisted in desiccation under vacuum conditions during the 24 h prior to the test, in order to avoid interferences due to evaporated water. Then, samples were placed on top of a carbon tape and a carbon bridge was created between the sample and the carbon tape. Finally, samples were coated with a thin layer of platinum. To obtain the images a voltage of 2 kV was employed.

### 3.7. Fourier Transform Infrared Spectroscopy (FTIR)

To study the degradation of PLA-PPy microfibers, the Fourier transform infrared spectroscopy (FTIR) analysis of samples belonging to three groups was carried out: PLA-PPy microfibers as control group; PLA-PPy microfibers degraded for 90 days in PB 0.1 M and stored at 37 °C; and PLA-PPy microfibers degraded at room temperature and atmospheric conditions for 180 days. FTIR analysis allows obtaining specific spectra for each sample, since each molecule or chemical bond generates transmittance and absorbance values at specific wavelengths. Subsequently, these spectra can be analysed to determine the bonds present in each sample after the degradation process. FTIR spectra of the samples were obtained using a Cary 630 FTIR (Agilent Technologies, Santa Clara, CA, USA) in the attenuated total reflection mode (ATR). The spectra resulted from averages of 24 scans at 4 cm^−1^ resolution, between 400 and 4000 cm^−1^. Four different samples (*n* = 4) of each group were studied, plotting the most representative curve for each one.

### 3.8. Thermogravimetric Analysis (TGA)

The thermogravimetric analysis (TGA) allows to study the thermal degradation of materials as well as their composition. In this case, to carry out the degradation study, the following experimental groups were studied: PLA-PPy microfibers as a control group; PLA-PPy microfibers after 90 days of degradation in PB 0.1 M at 37 °C; PLA-PPy microfibers after a degradation process of 180 days at room temperature and atmospheric conditions. A thermogravimetric analyser (TGA/SDTA 851 Mettler-Toledo operated using the METTLER: STARe Default DB V13.00 software, Barcelona, Spain) was employed. Samples with a mass of approximately 2 mg were processed, monitoring the mass loss while heating up to 800 °C at a rate of 10 °C/min under a positive nitrogen (N_2_) flow of 20 mL/min. Thermograms representing the mass loss of the sample as a function of temperature were obtained. Four different samples (*n* = 4) of each group were studied, plotting the most representative curve for each one.

### 3.9. Substrates Sanitization and Preconditioning

Before the seeding of cells, substrates were sanitized by 30 min of exposure to UV irradiation (UV lamp, wavelength 254 nm) on each side and by immersion in 70% ethanol (ET00021000, Scharlab, Madrid, Spain) for 3 washes of 10 min. Then the ethanol residues were removed by 4 washes of 10 min with ultrapure water (Milli-Q^®^, Sigma-Aldrich, Madrid, Spain). The substrates were then preconditioned by immersion in Dulbecco’s Modified Eagle Medium with a high glucose level (4.5 g/L) (21331020, Life Technologies, Madrid, Spain) supplemented with 10% foetal bovine serum (10270-106/A3381E, Life Technologies, Madrid, Spain) and 1% penicillin/streptomycin (15140122, Life Technologies, Madrid, Spain) and incubation at 37 °C for 24 h in a humidified atmosphere containing 5% CO_2_.

### 3.10. Seeding of SCs and DRG

Rat Schwan cells (SCs, P10301, Innoprot, Bizkaia, Spain), 8th cell passage, were grown at 37 °C, 5% CO_2_, in a SC medium (P60123, Innoprot, Bizkaia, Spain) until confluence. SCs were seeded with a density of 20,000 cells suspended in a 3 µL drop of the SC medium. For each well, 40,000 SCs were seeded (one drop of 3 µL at each end of the well). After 3 days of SC culture, dorsal root ganglia (DRG) from E8 chicken embryo were seeded (one at the centre of each well, *n* = 16 per group). To obtain the DRG, firstly chicken embryos were immersed in a Petri dish containing phosphate-buffered saline (PBS) 0.1 M to remove traces of the yolk and remaining fragments of the extra-embryonic tissue. Then, employing a dissecting microscope, the embryos were dissected using forceps to remove the tail, the peritoneal organs and the fat deposits in order to expose the spinal cord and the peripheral DRG in the sacro-lumbar region. The dorsal entry tracts of the sacro-lumbar DRG were exposed and DRG were carefully excised with ultrafine forceps by gently pulling away from the spine. Next, individual DRG explants were placed in the middle of every Millicell^®^ EZ slide well (Figure 4C). DRG were grown for 4 days in Ham F12 culture medium (11765054, ThermoFisher Scientific, Madrid, Spain) with 1% HEPES 1M (15630049, ThermoFisher Scientific, Madrid, Spain), 1% L-Glutamine 200 mM (25030024, Thermo Fisher Scientific, Madrid, Spain), 1% N2 supplement (17502048, Thermo Fisher Scientific, Madrid, Spain) and 1% penicillin/streptomycin (15140122, Life Technologies, Madrid, Spain) and 10 ng/mL of nerve growth factor (NGF, 13257019, Thermo Fisher Scientific, Madrid, Spain).

### 3.11. Electrical Stimulation Parameters

After 1 day of DRG culture, the electrical stimulation was applied with the following parameters: alternating current of 0.5 mA, frequency of 3 Hz, 8 h of continuous stimulation per day, and 3 days of total stimulation time. An AC voltage generator (PHYWE Function Generator, Gottingen, Germany) was used with the desired frequency and a sinusoidal signal. An ammeter (FLUKE 45 Dual Display Multimeter, Madrid, Spain) was connected in series to set the circulating current.

### 3.12. Immunostaining of SCs and DRG

After cell culture, the cell medium was removed and samples were washed in PB 0.1 M and fixed with 4% paraformaldehyde (PFA, 47608, Sigma-Aldrich, Madrid, Spain) for 20 min at room temperature. After cell fixation, 3 washes of 10 min with PB 0.1 M were performed to remove PFA residues. Then, the non-specific bindings were blocked, and the cell membrane was permeabilized by immersion in a blocking buffer composed of PB 0.1 M with 3% bovine serum albumin (BSA, A7906, Sigma-Aldrich, Madrid, Spain) and 0.1% Tween20 (P1379, Sigma-Aldrich, Madrid, Spain) for 1 h at room temperature. SCs were stained with rabbit monoclonal anti-S100β antibody (ab52642, Abcam, Cambridge, UK, 1/200 dilution) and DRG were stained with Neuron-specific β III Tubulin Antibody (MAB1195, R&D Systems, Madrid, Spain, 1/100 dilution). Samples were incubated with primary antibodies at 4 °C overnight. Next, secondary antibodies, goat anti-rabbit IgG Alexa Fluor^®^ 555 (A-21429, Thermo Fisher Scientific, Madrid, Spain, 1/200 dilution) and goat anti-mouse IgG Alexa Fluor^®^ 488 (A28175, ThermoFisher Scientific, Madrid, Spain, 1/200 dilution) were incubated for another 2 h at room temperature in the darkness. Afterward, samples were incubated with 4′,6-diamidino-2-phenylindole (DAPI, D9564, Sigma-Aldrich, Madrid, Spain, 1/1000 dilution) for 10 min to mark the cell nuclei. The imaging was performed using a confocal microscope (LEICA TCS SP5, Leica microsystems, Wetzlar, Germany) and a fluorescence microscope (Leica DM5000, Leica microsystems, Germany, 5× magnification). For the axonal quantification images, the following parameters were used in the Leica DM5000 fluorescence microscope: exposure of 1.2 s, gain of 8.1×, saturation of 1.15 and gamma of 0.97.

### 3.13. Quantification of Axonal Extension and Axonal Sprouting

To quantify the axonal extension, two different parameters were obtained. On the one hand, the maximum length of axons was measured as the distance between the edge of the DRG body and the end of the longest axon, as the maximum, sum and average of left and right sides of the DRG body. On the other hand, the area of axons was obtained by subtracting the area of the DRG body from the total area of the DRG. Both parameters were measured with the ImageJ/FIJI image processing software [101].

### 3.14. RNA Extraction and cDNA Synthesis

To evaluate the effect of the electrical stimulation on the gene expression of the co-culture of SCs and DRG, the total RNA was extracted employing QIAzol lysis reagent (79306, Qiagen, Madrid, Spain). First, 350 μL of QIAzol were added to each sample, and then they were incubated at room temperature for 5 min. The supernatant was transferred, 70 μL of chloroform were added, and the samples were incubated at room temperature for 5 min. Then, the samples were centrifuged (15 min, 14,000 rpm, 4 °C). The aqueous layer was mixed with 175 μL of isopropanol and kept at room temperature for 5 min. Samples were centrifuged (10 min, 14,000 rpm, 4 °C). The pellet was resuspended in 350 μL of 75% ethanol and the samples were centrifuged (5 min, 14,000 rpm, 4 °C). The ethanol was evaporated at room temperature. The resulting pellet was dissolved in 30 μL of RNAse free water. RNA concentration, integrity and quality were measured with a Q3000 UV Spectrophotometer (Quawell, San Jose, CA, USA).

For cDNA synthesis, approximately 500 ng of total RNA was converted into cDNA using the Maxima First Strand cDNA Synthesis Kit for RT-qPCR with dsDNase (K1671, ThermoFisher Scientific, Madrid, Spain) in a reaction volume of 20 μL. The reaction was conducted with the following conditions: 25 °C for 10 min, 55 °C for 20 min, 85 °C for 5 min and a final hold at 4 °C. The resulting cDNA quality and concentration were measured using a Q3000 UV Spectrophotometer (Quawell, San Jose, USA) and it was stored at −80 °C until further analysis. The experiment was carried out in quadruplicate (*n* = 4).

### 3.15. Quantitative Real-Time PCR

Quantitative real-time polymerase chain reactions (qRT-PCR) were carried out on 96-well plates (Applied Biosystems^®^, ThermoFisher Scientific, Madrid, Spain) with each sample represented by the gene of interest and one housekeeping gene (Glyceraldehyde Phosphate Dehydrogenase, GAPDH). The primers for each gene were designed with the PRIMER3plus software (www.bioinformatics.nl accessed on 7 February 2022) based on specific DNA sequences from NCBI (https://www.ncbi.nlm.nih.gov/nucleotide/ accessed on 7 February 2022), which were purchased from ThermoFischer Scientific (Madrid, Spain). The primer sequences of the target genes are shown in Table 2. All the primer sequences of target genes were species specific, so no amplification of chicken genes was obtained for samples containing only rat RNA and no amplification of rat genes was obtained for samples containing only chicken RNA. This allowed us to study genes of a specific species in a co-culture of two different species (chicken and rat), allowing us to study separately the biomarkers of DRG-related cells and of pre-seeded SCs. Individual reactions contained 2 μL of cDNA with a concentration of 1 ng/µL, 1 μL of specific primers (forward and reverse at a concentration of 10 μM), 5 μL of SYBR™ PowerUp™ SYBR™ Green Master Mix (15350929, Fisher Scientific, Madrid, Spain) and 1 μL of RNAse-free water in a final total volume of 10 μL. Amplification efficiency was analysed before qRT-PCR to optimize measurements. Reactions were carried out in a StepOne Plus™ Real-Time PCR System (Applied Biosystems^®^, ThermoFisher Scientific, Madrid, Spain) with a holding stage of 50 °C for 2 min and 95 °C for 2 min, followed by a cycling stage of 40 cycles of 95 °C for 15 s and 60 °C for 1 min. The data were obtained using the StepOne Plus™ Software 2.3 (Applied Biosystems^®^, ThermoFisher Scientific, Madrid, Spain). Fold changes were calculated using the 2^−ΔΔCt^ method. Four biological replicates and four technical replicates were measured for each sample. The data were represented as the relative expression for the electrically stimulated group with respect to the control group for each one of the genes.

### 3.16. Statistical Analysis

Results are expressed as the mean ± standard error of the mean (SEM). The statistical analysis of the results was performed with GraphPad Prism^®^ v.6 software (GraphPad Software, San Diego, CA, USA) in order to reveal significant differences between conditions. The two-sided, unpaired *t*-test was applied (confidence level of 95%). Statistically significant differences are indicated by *, **, *** or ****, indicating a *p*-value below 0.05, 0.01, 0.001 or 0.0001, respectively.

## 4. Conclusions

The experimental data proves the ability to obtain electrically conductive, highly aligned and biocompatible PLA-PPy-Au substrates based on microfibers, which greatly expands the potential of this composite material to be used for neural tissue engineering and implantable-device applications. A high axonal extension and axonal sprouting is observed for DRG cultured on the PLA-PPy-Au microfiber substrates, reaching an increment of 19.2% in the maximum length of the axons and 40% in the area of the axons when electrically stimulated. In addition, the electrically stimulated group presented a higher expression of neuronal-, Schwann cell-, proliferation- and neurotrophin-related biomarkers, also confirming that this group has stronger nerve regeneration potential. In conclusion, the usefulness of the substrate to increase and accelerate axonal growth when subjected to an exogenous electrical stimulus is proved.

## Figures and Tables

**Figure 1 ijms-23-06362-f001:**
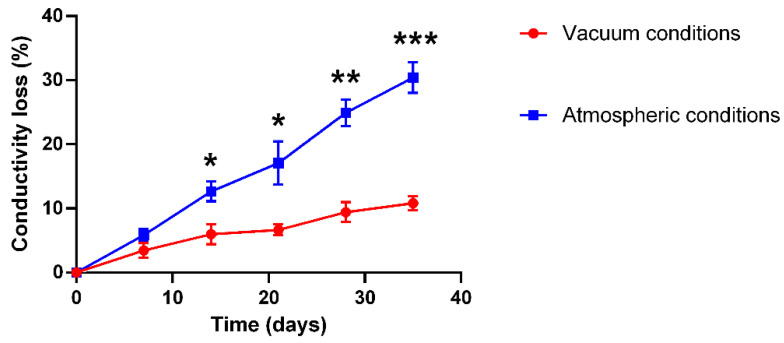
Conductivity loss of PLA-PPy microfibers stored under vacuum conditions and under atmospheric conditions for 35 days. The samples stored under atmospheric conditions suffered a higher conductivity loss than the samples stored under vacuum conditions. Statistically significant differences are indicated by *, ** or ***, indicating a *p*-value below 0.05, 0.01 or 0.001, respectively.

**Figure 2 ijms-23-06362-f002:**
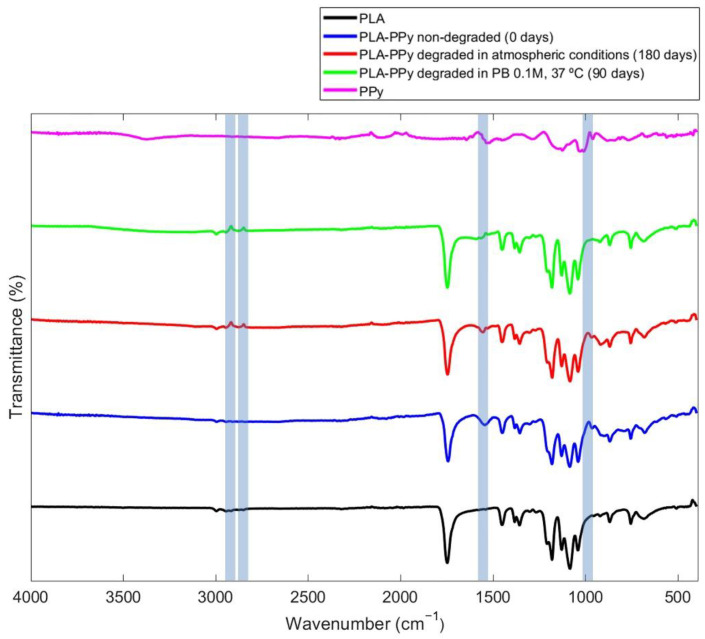
FTIR spectra of PLA microfibers (PLA), non-degraded PLA-PPy microfibers, PLA-PPy microfibers degraded in atmospheric conditions for 180 days, PLA-PPy microfibers degraded in PB 0.1 M at 37 °C for 90 days and pure PPy (PPy). The vertical shaded areas are highlighting the differential characteristic peaks that are studied.

**Figure 3 ijms-23-06362-f003:**
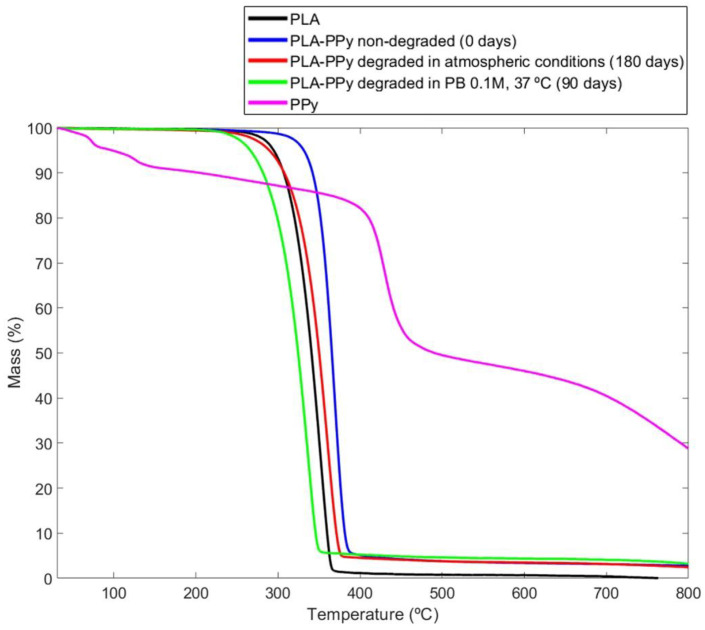
Thermogravimetric analysis (TGA) of PLA microfibers (PLA), non-degraded PLA-PPy microfibers, PLA-PPy microfibers degraded in atmospheric conditions for 180 days, PLA-PPy microfibers degraded in PB 0.1 M at 37 °C for 90 days and pure PPy (PPy).

**Figure 4 ijms-23-06362-f004:**
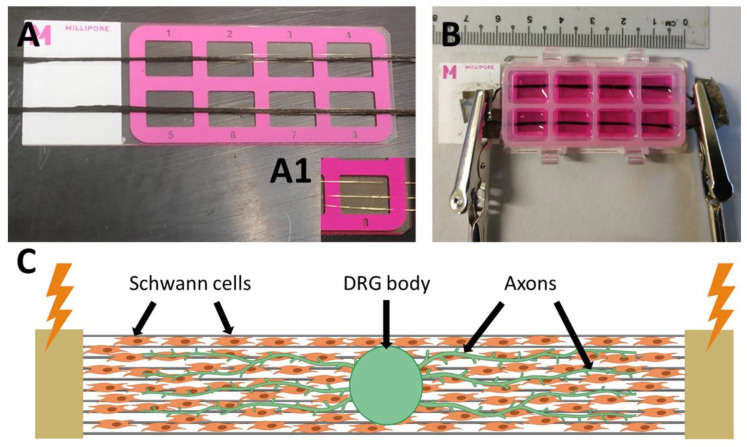
Images of the bioreactor for electrical stimulation. (**A**) Image of the PLA-PPy-Au microfiber lane placed on the top of the slide. The gold microfibers are placed below the PLA-PPy microfibers, as can be appreciated in insert A1. (**B**) Image of the fully assembled device, showing the electrodes placed at the extremes of the microfibers. Two PLA-PPy-Au substrates are placed between the slide and the wells, one per row, and connected in parallel to the electrodes avoiding contact with the culture medium. (**C**) Scheme showing the microfiber lane corresponding to one of the wells with the coverage of pre-seeded Schwann cells, the DRG body containing the somas of the neurons and the axons sprouting from the DRG body in an aligned way following the direction of the microfibers.

**Figure 5 ijms-23-06362-f005:**
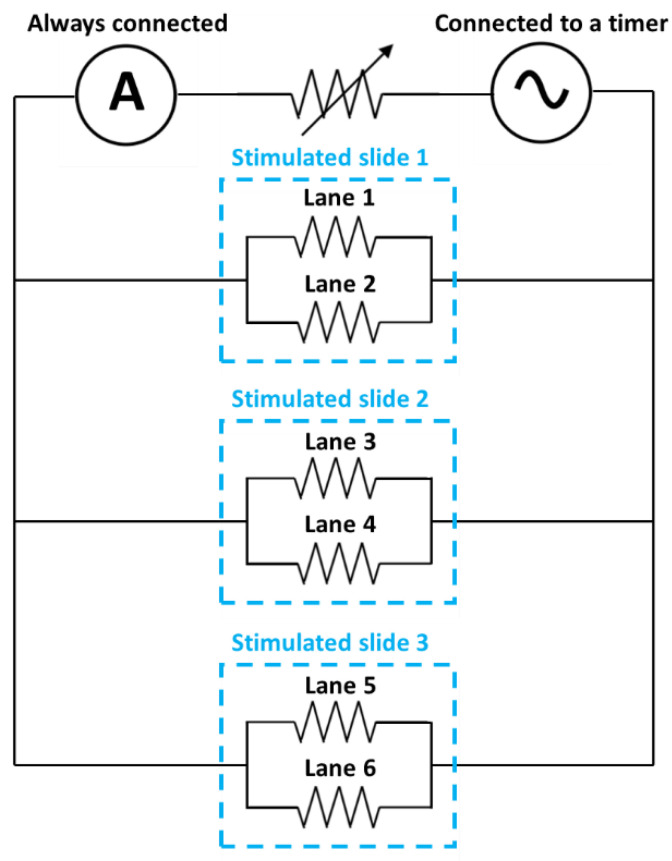
Electrical diagram of the different elements used. Six PLA-PPy-Au lanes were connected in parallel using three different slide chambers so that the same current circulated through each lane. An AC generator at a given frequency and with a sinusoidal signal fed the circuit and the circulating current was measured in real time with an ammeter connected in series. A potentiometer was also connected in series to raise the total electrical resistance of the circuit in order to use a voltage that was not excessively low for the voltage generator.

**Figure 6 ijms-23-06362-f006:**
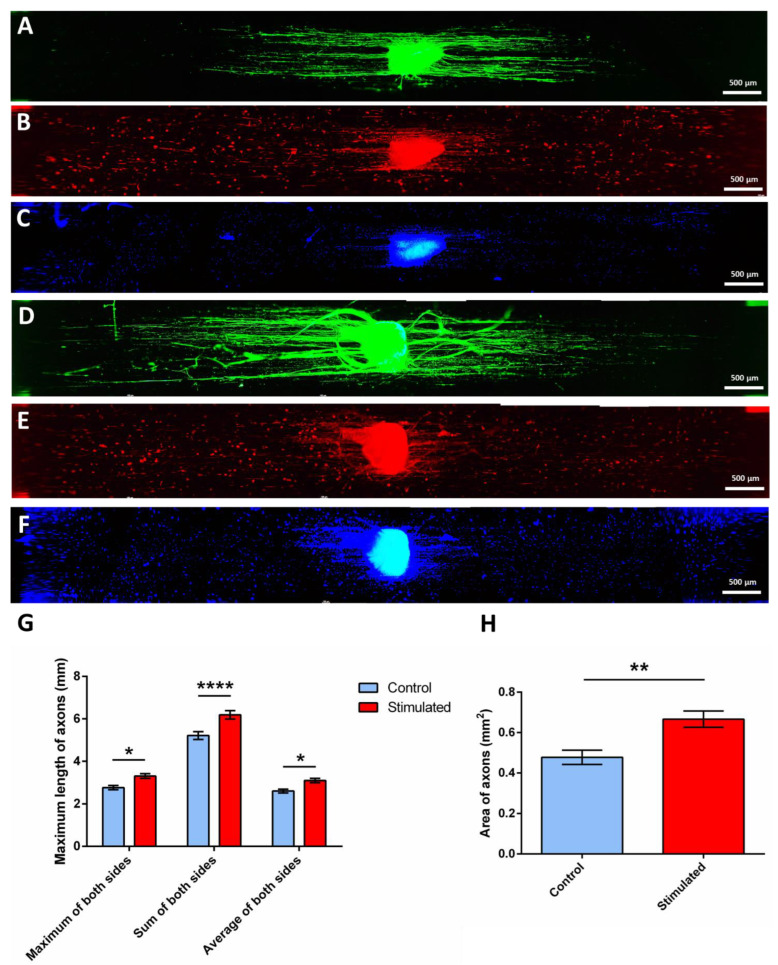
Example of control (**A**–**C**) and electrically stimulated (**D**–**F**) samples. Axons marked with β III Tubulin are shown in green colour (**A**,**D**), Schwann cells marked with S100β are shown in red colour (**B**,**E**) and the cell nuclei marked with DAPI are shown in blue colour (**C**,**F**). It can be observed the increase in the length and in the total area occupied by axons for the electrically stimulated sample. The presence of pre-seeded Schwann cells also can be confirmed since they are present far away from the axons. (**G**) Quantification of axonal extension through the measurement of the maximum length of axons as the maximum, sum and average at both sides of the DRG body. (**H**) Quantification of axonal sprouting through the measurement of the area occupied by axons. The data in G and H are presented as the mean ± SEM (*n* = 16). Statistically significant differences are indicated by *, ** or ****, indicating a *p*-value below 0.05, 0.01 or 0.0001, respectively.

**Figure 7 ijms-23-06362-f007:**
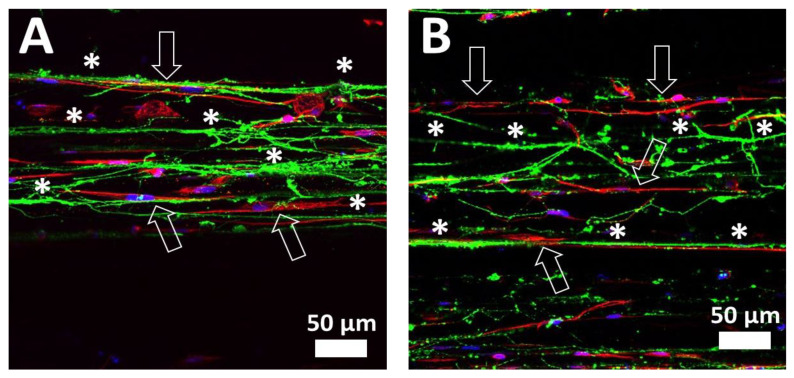
Confocal images of control (**A**) and electrically stimulated (**B**) samples. This confocal images in a region far from the DRG body allows to observe the axons (Neuron-specific β III Tubulin antibody in green colour) and the Schwann cells (S100β antibody in red colour), as well as the cell nuclei (DAPI in blue colour). It can be observed how the axons grow in a highly aligned manner in the direction of the PLA-PPy microfibers (marked with *). Likewise, it is also observed that the cytoskeletons of Schwann cells are elongated in the direction of PLA-PPy microfibers (marked with arrows), accompanying the axons.

**Figure 8 ijms-23-06362-f008:**
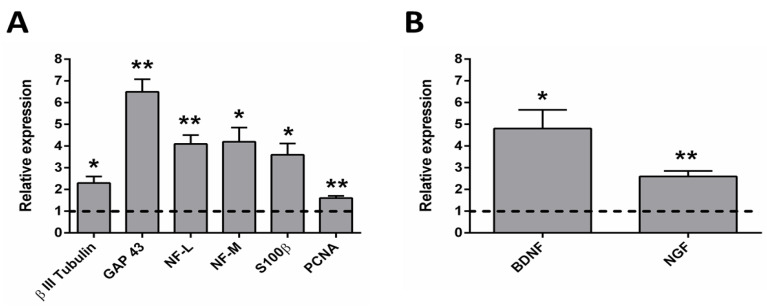
Relative expression levels of the genes studied by RT-qPCR. (**A**) Chicken genes related with axonal growth (β III Tubulin, GAP 43, neurofilament low chain (NF-L) and neurofilament medium chain (NF-M)), with Schwann cells (S100β) and with cell proliferation (PCNA), normalized to chicken GAPDH. (**B**) Rat genes related with brain-derived neurotrophic factor (BDNF) and nerve growth factor (NGF), normalized to rat GAPDH. Statistically significant differences are indicated by * or **, indicating a *p*-value below 0.05 or 0.01, respectively.

**Table 1 ijms-23-06362-t001:** Stimulation parameters: different electric currents and frequencies were applied to the cultures, evaluating their effect on axonal growth. The total stimulation time (3 days), the continuous stimulation time per day (8 h), the culture duration (4 days), the pre-seeding of SCs (40,000 cells/well), the origin of DRG (chicken embryos) and the age of DRG (E8) were kept constant.

Experiment Number	Current per Lane (mA)	Voltage per Lane (mV)	Frequency (Hz)	Effect on Axonal Growth
#1	3	30.6	250	Adverse effect
#2	1	10.2	250	Adverse effect
#3	0.5	5.1	50	No effect
#4	0.5	5.1	3	Favourable effect

**Table 2 ijms-23-06362-t002:** Primer sequences for the chicken and rat target genes that were studied.

Species	Gen	Primer Forward	Primer Reverse
Chicken (Gallus Gallus)	β III Tubulin	TCCTCTCACAAGTACGTGCC	CCCCGCTCTGACCGAAAAT
	NF-L	AAGACGCTGGAGATCGAAGC	CACCTTCCAGCAGTTTCCTGT
	NF-M	CACCACCTATCAGGACACGAT	GGGTCCAGTGATGCTTCCAG
	GAP 43	CATAAGGCAGCCACCAAAAT	CGGAAGCCTCACTCTCTTTG
	S100β	TGCTTGCCATGAGTTCTTTG	GCACTGTCCAAGAGGCTTTC
	PCNA	GAGACCTCAGCCACATTGGT	AGTCAGCTGGACTGGCTCAT
	GAPDH	AGTCAACGGATTTGGCCGTA	ACAGTGCCCTTGAAGTGTCC
Rat (Rattus norvegicus)	BDNF	GCGGCAGATAAAAAGACTGC	GTAGTTCGGCATTGCGAGTT
	NGF	TGATCGGCGTACAGGCAGA	GAGGGCTGTGTCAAGGGAAT
	GAPDH	AGACAGCCGCATCTTCTTGT	GACCAGCTTCCCATTCTCAG

## Data Availability

All the data generated in this research are included in the manuscript.

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
