# Peer review of "Electrical Stimulation Increases Axonal Growth from Dorsal Root Ganglia Co-Cultured with Schwann Cells in Highly Aligned PLA-PPy-Au Microfiber Substrates"

_ijms, 2022, doi:10.3390/ijms23126362_

Round 1
Reviewer 1 Report
Roca et al presented clear evidence about using electrical stimulation to promote the axon growth of DRGs on PLA-PPy-Au microfibers. The authors studied the stability of the material and the underlying mechanism for improved axon growth. In general, the manuscript is well organised and easy to follow. A few questions:
1. Can authors elaborate the contribution of this work to the field? Electrical stimulation has already been demonstrated before for promoting DRGs on other microfibers. For example: https://www.ncbi.nlm.nih.gov/pmc/articles/PMC4873603/. What is the novelty of this work? Have the authors presented any new underlying mechanism? How does their material compare to other microfibers?
2. The authors showed clear evidence that the material degrade after the stability tests. Do authors have any suggestion how to improve the stability? If the material is unstable, does this indicate that the material may not be suitable for the application proposed in this work?
3. The stability tests were performed without gold coating. Is this reasonable?
4. The authors mentioned in Section 3.2 that DRG cannot adhere to a substrate formed by gold microfibers. Can authors please provide ref or data?
5. I suggest the authors to include the SEM images of their substrate in the manuscript to help readers understand their material.
Author Response
Reviewer 1
Roca et al presented clear evidence about using electrical stimulation to promote the axon growth of DRGs on PLA-PPy-Au microfibers. The authors studied the stability of the material and the underlying mechanism for improved axon growth. In general, the manuscript is well organised and easy to follow. A few questions:
- Can authors elaborate the contribution of this work to the field? Electrical stimulation has already been demonstrated before for promoting DRGs on other microfibers.
For example: https://www.ncbi.nlm.nih.gov/pmc/articles/PMC4873603/. What is the novelty of this work? Have the authors presented any new underlying mechanism? How does their material compare to other microfibers?
Our work presents various novelties within the field of nervous tissue engineering. On the one hand, the diameter of the electroconductive fibers is 10 microns, which is not the usual size used in this type of studies. Since fiber substrates obtained by means of the electrospinning technique are normally used, nanofibers with a diameter below 1 micron are commonly employed [1,2]. We previously observed that the use of microfiber substrates led to an increased axonal extension and axonal sprouting than the use of nanofiber substrates, since on microfibers the cell mobility was much less hindered than on nanofibers, and the cell-cell interactions were favoured, resulting in a higher motility of SC that accelerated their migration speed from the DRG body, which is directly related to the axonal extension [3]. In addition, the PPy coating of the PLA microfibers resulted in an additional beneficial effect on axonal growth, possibly produced by the greater surface roughness that favoured the adhesion of DRG [3].On the other hand, most studies of electrical stimulation of DRG or isolated neurons apply stimulation by introducing electrodes into the culture medium, commonly using non-electroconductive substrates, since the current circulates through the culture medium and not through the culture substrate [4–6]. Furthermore, the few studies that apply the electrical stimulation through an electroconductive substrate do not employ microfibers [7–9]. Therefore, the fact that we use a substrate based on electroconductive microfibers which is connected outside the culture well (without medium-electrode contact) so that the current circulates mainly through the culture substrate and not through the culture medium represents a novelty in this field of study. In addition, the combination of PPy coated microfibers with gold microfibers is also a novelty and, with this disposition, an electric field around the gold microfibers is created, generating an additional type of stimulation.
Another novelty of this study is that the entire DRG is stimulated (not just isolated neurons as is done in most studies) in co-culture with SCs, so it is a co-stimulation of the DRG and the SCs. Finally, the type of current typically used in the stimulation of neurons or DRG is direct current (DC), introducing salt bridges to minimize the exposure of cells to changes in pH due to the accumulation of ionic charges in the culture medium [4,6]. In this study we apply alternating current (AC) in order to avoid the accumulation of ionic charges of the medium and thus avoiding having to use salt bridges. AC stimulation has been applied in some studies where the current circulated through the culture medium and not through a substrate based on electroconductive microfibers [7,10].
Regarding the molecular mechanisms, we have studied the main proteins related with axonal growth: β III Tubulin related with neural development, GAP 43 related with growth cones during axonal regeneration, and neurofilament related with the integrity of neuronal cytoarchitecture, which were more expressed for the electrically stimulated group. Also, two of the main neurotrophic factors (BDNF and NGF) have been studied, which were also more expressed for the electrically stimulated group. These are the main biomarkers that are related with axonal growth and regeneration, and they are not usually studied in this kind of papers. Also, the fact of having a co-culture of two different species (chicken and rat) allows us to study separately the biomarkers of DRG-related cells and the pre-seeded SCs.
These comments have been introduced in the discussion of the results in the manuscript of the paper, highlighted in yellow colour.
- The authors showed clear evidence that the material degrade after the stability tests. Do authors have any suggestion how to improve the stability? If the material is unstable, does this indicate that the material may not be suitable for the application proposed in this work?
As indicated in the manuscript, in order to increase the stability of the PPy coating, the microfibers were stored under vacuum conditions. In this way it is achieved that after 35 days the loss of conductivity decreases from 30% to 11%, which is a small loss in a relatively long time. It is important to highlight that this is an in vitro study with ideal conditions. Regarding the in vivo situation after its implantation, the regeneration of nerves takes several months and we expect a loss of conductivity of PPy in this situation, but the PPy performance should be effective at least during the first weeks that are the more critical ones. Despite this should be further studied in future experiments, many groups have already reported the biodegradation properties of PLA-PPy blends [2,11–13]. PLA-PPy composites showed a good electroconductivity for long time in a simulated physiological solution (15% of the conductivity remained after 1000 h) and the in vivo biodegradation of the composites showed the non-toxicity of degradation products [11–14]. Therefore, we expect that the in vivo degradation of the PLA-PPy substrates will not lead to a problematic immune response. In addition, we have proved that the PPy coverage does not prevent the degradation of PLA microfibers, which is an advantage for the in vivo application since the PLA-PPy microfibers will be satisfactorily resorbed.
These comments have been introduced in the discussion of the results in the manuscript of the paper, highlighted in yellow colour.
- The stability tests were performed without gold coating. Is this reasonable?
Regarding gold microfibers, gold is considered chemically inert with a high stability and its surface shows low chemisorption capacity [15]. This causes gold to exhibit low chemical reactivity, little catalytic activity, high stability against corrosion and oxidation [16–18]. Therefore, clinically, gold presents biocompatibility when in contact with body fluids and tissues, adequate characteristics to avoid immunogenic reactions [15,17]. Furthermore, as it is a metal, it has excellent conductivity and has been used in electrodes for electrical neurostimulation [19,20]. Gold is highly stable in vivo, and its conductivity should not vary during the time needed for nerve regeneration. In addition, on our study gold is not a coating of the PLA-PPy microfibers, but is incorporated in the form of independent microfibers, so there should be no chemical interaction between PPy and gold microfibers. For these reasons, the stability and degradation of gold microfibers has not been studied.
These comments have been introduced in the discussion of the results in the manuscript of the paper, highlighted in yellow colour.
- The authors mentioned in Section 3.2 that DRG cannot adhere to a substrate formed by gold microfibers. Can authors please provide ref or data?
We have included a supplementary figure (Figure S3) of DRG seeded on gold substrates, showing their poor adherence to the substrate so there is not axonal extension after 4 days of cell culture. We attribute this to the fact that the electrochemical nature of bare gold is very neutral, which is why it does not favour the adsorption of proteins on gold during the pre-conditioning process of the substrate and, therefore, greatly hinders the adhesion of DRG to gold. Anchorage-dependent cells make integrin-mediated cell adhesion which has not affinity for bare gold, so the gold surface must be modified, for example with thiol-functionalized RGD peptide, in order to achieve cell adhesion to gold [21].
These comments have been introduced in the discussion of the results in the manuscript of the paper, highlighted in yellow colour.
Figure S3. Chicken DRG (marked with arrows) seeded on top of a gold substrate (marked with *) after 4 days of cell culture. Immunostaining with β III Tubulin is observed in green colour. As can be observed, there is no axonal extension from the DRG bodies, indicating a poor adhesion of DRG bodies to the gold substrate.
- I suggest the authors to include the SEM images of their substrate in the manuscript to help readers understand their material.
SEM images have been included in the manuscript (Figure S1), as suggested, showing the PLA-PPy microfibers. The corresponding section on materials and methods has also been added, indicating how the SEM images were obtained. The following text have been added to the discussion section highlighted in yellow colour: “In this work, substrates made of highly aligned PLA microfibers were coated with the electrically conductive polymer polypyrrole (PPy) via in situ polymerization with a mass fraction of PPy of 3.5% (Figure S1). In previous studies [22,23], it was observed by scanning electron microscopy that these PLA-PPy substrates showed a continuous and homogeneous PPy coating with a fine grain texture. Only small PPy aggregates were observed and most of them disappeared with repeated washing, so this coating protocol with PPy was applied.”
Figure S1. A: PLA microfiber lane previously to the PPy coating. Microfibers with a diameter of 10 µm were employed. B: Detail of the PLA microfibers previously to the PPy coating. C: PLA-PPy microfiber lane. D: Detail of the PLA-PPy microfibers. A homogeneous PPy coating can be observed, with a fine grain texture. Only some aggregates (marked with *) were present and most of them disappeared with repeated washing.
References
- Zou, Y.; Qin, J.; Huang, Z.; Yin, G.; Pu, X.; He, D. Fabrication of Aligned Conducting PPy-PLLA Fiber Films and Their Electrically Controlled Guidance and Orientation for Neurites. ACS Appl. Mater. Interfaces 2016, 8, 12576–12582, doi:10.1021/ACSAMI.6B00957/SUPPL_FILE/AM6B00957_SI_001.PDF.
- Lee, J.Y.; Bashur, C.A.; Goldstein, A.S.; Schmidt, C.E. Polypyrrole-Coated Electrospun PLGA Nanofibers for Neural Tissue Applications. Biomaterials 2009, 30, 4325–4335, doi:10.1016/J.BIOMATERIALS.2009.04.042.
- Gisbert Roca, F.; Más Estellés, J.; Monleón Pradas, M.; Martínez-Ramos, C. Axonal Extension from Dorsal Root Ganglia on Fibrillar and Highly Aligned Poly(Lactic Acid)-Polypyrrole Substrates Obtained by Two Different Techniques: Electrospun Nanofibres and Extruded Microfibres. Int. J. Biol. Macromol. 2020, 163, 1959–1969, doi:10.1016/J.IJBIOMAC.2020.09.181.
- Koppes, A.N.; Zaccor, N.W.; Rivet, C.J.; Williams, L.A.; Piselli, J.M.; Gilbert, R.J.; Thompson, D.M. Neurite Outgrowth On Electrospun PLLA Fibers Is Enhanced By Exogenous Electrical Stimulation. J. Neural Eng. 2014, 11, 046002, doi:10.1088/1741-2560/11/4/046002.
- Kumar, P.J.; Adams, R.D.; Harkins, A.B.; Engeberg, E.D.; Willits, R.K. Stimulation Frequency Alters the Dorsal Root Ganglion Neurite Growth and Directionality In Vitro. IEEE Trans. Biomed. Eng. 2016, 63, 1257–1268, doi:10.1109/TBME.2015.2492998.
- Bertucci, C.; Koppes, R.; Dumont, C.; Koppes, A. Neural Responses to Electrical Stimulation in 2D and 3D in Vitro Environments. Brain Res. Bull. 2019, 152, 265–284, doi:10.1016/j.brainresbull.2019.07.016.
- Quigley, A.F.; Razal, J.M.; Thompson, B.C.; Moulton, S.E.; Kita, M.; Kennedy, E.L.; Clark, G.M.; Wallace, G.G.; Kapsa, R.M.I. A Conducting-Polymer Platform with Biodegradable Fibers for Stimulation and Guidance of Axonal Growth. Adv. Mater. 2009, 21, 4393–4397, doi:10.1002/adma.200901165.
- Imaninezhad, M.; Pemberton, K.; Xu, F.; Kalinowski, K.; Bera, R.; Zustiak, S.P. Directed and Enhanced Neurite Outgrowth Following Exogenous Electrical Stimulation on Carbon Nanotube-Hydrogel Composites. J. Neural Eng. 2018, 15, doi:10.1088/1741-2552/aad65b.
- Zhou, Z.; Liu, X.; Wu, W.; Park, S.; Miller, A.L.; Terzic, A.; Lu, L. Effective Nerve Cell Modulation by Electrical Stimulation of Carbon Nanotube Embedded Conductive Polymeric Scaffolds. Biomater. Sci. 2018, 6, 2375–2385, doi:10.1039/c8bm00553b.
- Chang, Y.J.; Hsu, C.M.; Lin, C.H.; Lu, M.S.C.; Chen, L. Electrical Stimulation Promotes Nerve Growth Factor-Induced Neurite Outgrowth and Signaling. Biochim. Biophys. Acta - Gen. Subj. 2013, 1830, 4130–4136, doi:10.1016/j.bbagen.2013.04.007.
- Wang, Z.; Roberge, C.; Dao, L.H.; Wan, Y.; Shi, G.; Rouabhia, M.; Guidoin, R.; Zhang, Z. In Vivo Evaluation of a Novel Electrically Conductive Polypyrrole/Poly(D,L- Lactide) Composite and Polypyrrole-Coated Poly(D,L-Lactide-Co-Glycolide) Membranes. J. Biomed. Mater. Res. - Part A 2004, 70, 28–38, doi:10.1002/jbm.a.30047.
- Wang, Z.; Roberge, C.; Wan, Y.; Dao, L.H.; Guidoin, R.; Zhang, Z. A Biodegradable Electrical Bioconductor Made of Polypyrrole Nanoparticle/Poly(D,L-Lactide) Composite: A Preliminary in Vitro Biostability Study. J. Biomed. Mater. Res. - Part A 2003, 66, 738–746, doi:10.1002/jbm.a.10037.
- Huang, Z.B.; Yin, G.F.; Liao, X.M.; Gu, J.W. Conducting Polypyrrole in Tissue Engineering Applications. Front. Mater. Sci. 2014, 8, 39–45, doi:10.1007/s11706-014-0238-8.
- Wang, X.; Gu, X.; Yuan, C.; Chen, S.; Zhang, P.; Zhang, T.; Yao, J.; Chen, F.; Chen, G. Evaluation of Biocompatibility of Polypyrrole in Vitro and in Vivo. J. Biomed. Mater. Res. - Part A 2004, 68, 411–422, doi:10.1002/jbm.a.20065.
- Alex, S.; Tiwari, A. Functionalized Gold Nanoparticles: Synthesis, Properties and Applications-A Review. J. Nanosci. Nanotechnol. 2015, 15, 1869–1894, doi:10.1166/jnn.2015.9718.
- Vigneron, F.; Caps, V. Evolution in the Chemical Making of Gold Oxidation Catalysts. Comptes Rendus Chim. 2016, 19, 192–198, doi:10.1016/j.crci.2015.11.015.
- Das, M.; Shim, K.H.; An, S.S.A.; Yi, D.K. Review on Gold Nanoparticles and Their Applications. Toxicol. Environ. Health Sci. 2011, 3, 193–205, doi:10.1007/s13530-011-0109-y.
- Tiwari, P.M.; Vig, K.; Dennis, V.A.; Singh, S.R. Functionalized Gold Nanoparticles and Their Biomedical Applications. Nanomater. 2011, Vol. 1, Pages 31-63 2011, 1, 31–63, doi:10.3390/NANO1010031.
- Domínguez-Bajo, A.; Rosa, J.M.; González-Mayorga, A.; Rodilla, B.L.; Arché-Núñez, A.; Benayas, E.; Ocón, P.; Pérez, L.; Camarero, J.; Miranda, R.; et al. Nanostructured Gold Electrodes Promote Neural Maturation and Network Connectivity. Biomaterials 2021, 279, doi:10.1016/j.biomaterials.2021.121186.
- Lienemann, S.; Zötterman, J.; Farnebo, S.; Tybrandt, K. Stretchable Gold Nanowire-Based Cuff Electrodes for Low-Voltage Peripheral Nerve Stimulation. J. Neural Eng. 2021, 18, doi:10.1088/1741-2552/abfebb.
- Yoon, S.H.; Mofrad, M.R.K. Cell Adhesion and Detachment on Gold Surfaces Modified with a Thiol-Functionalized RGD Peptide. Biomaterials 2011, 32, 7286–7296, doi:10.1016/j.biomaterials.2011.05.077.
- Gisbert Roca, F.; Más Estellés, J.; Monleón Pradas, M.; Martínez-Ramos, C. Axonal Extension from Dorsal Root Ganglia on Fibrillar and Highly Aligned Poly(Lactic Acid)-Polypyrrole Substrates Obtained by Two Different Techniques: Electrospun Nanofibres and Extruded Microfibres. Int. J. Biol. Macromol. 2020, 163, 1959–1969, doi:10.1016/j.ijbiomac.2020.09.181.
- Gisbert Roca, F.; André, F.M.; Más Estellés, J.; Monleón Pradas, M.; Mir, L.M.; Martínez-Ramos, C. BDNF-Gene Transfected Schwann Cell-Assisted Axonal Extension and Sprouting on New PLA–PPy Microfiber Substrates. Macromol. Biosci. 2021, 21, 1–13, doi:10.1002/mabi.202000391.

Reviewer 2 Report
Superb manuscript. Can the authors relate the effective stimulation parameters to what may go on naturally....what activities (frequency of firing etc) do dorsal root afferents have in the body upon natural activation?
Author Response
Reviewer 2
Superb manuscript. Can the authors relate the effective stimulation parameters to what may go on naturally....what activities (frequency of firing etc) do dorsal root afferents have in the body upon natural activation?
Several studies of neurostimulation of DRG have observed that low frequencies (<20 Hz) have a beneficial effect on pain relief, inducing the dorsal horn of the spinal cord inhibition via the activation of low threshold mechanoreceptors and the release of endorphins and dynorphins [1–4]. Therefore, we could say that the applied stimulation frequency of 3 Hz is within the physiological range of DRG activation. Regarding the applied current, a current of 1 mA has been applied also for neural activation during DRG stimulation [1], so the applied frequency of 0.5 mA is also in the physiological range of DRG activation. These comments have been introduced in the discussion of the results in the manuscript of the paper, highlighted in yellow colour.
References
- Graham, R.D.; Bruns, T.M.; Duan, B.; Lempka, S.F. The Effect of Clinically Controllable Factors on Neural Activation During Dorsal Root Ganglion Stimulation. Neuromodulation 2021, 24, 655–671, doi:10.1111/ner.13211.
- Vuka, I.; Vučić, K.; Repić, T.; Ferhatović Hamzić, L.; Sapunar, D.; Puljak, L. Electrical Stimulation of Dorsal Root Ganglion in the Context of Pain: A Systematic Review of In Vitro and In Vivo Animal Model Studies. Neuromodulation 2018, 21, 213–224.
- Bauman, M.J.; Bruns, T.M.; Wagenaar, J.B.; Gaunt, R.A.; Weber, D.J. Online Feedback Control of Functional Electrical Stimulation Using Dorsal Root Ganglia Recordings. Conf. Proc. IEEE Eng. Med. Biol. Soc. 2011, 2011, 7246, doi:10.1109/IEMBS.2011.6091831.
- Franken, G.; Douven, P.; Debets, J.; Joosten, E.A.J. Conventional Dorsal Root Ganglion Stimulation in an Experimental Model of Painful Diabetic Peripheral Neuropathy: A Quantitative Immunocytochemical Analysis of Intracellular γ-Aminobutyric Acid in Dorsal Root Ganglion Neurons. Neuromodulation 2021, 24, 639–645, doi:10.1111/ner.13398.

Round 2
Reviewer 1 Report
The authors have addressed all my concerns. Thank you.